# Anamnesic Neural Differential Equations with Orthogonal Polynomials Projections

**Edward De Brouwer**
ESAT-STADIUS
KU Leuven
Leuven, Belgium
edward.debrouwer@kuleuven.be

**Rahul G. Krishnan**
Department of Computer Science
University of Toronto
Toronto, Canada
rahulgk@cs.toronto.edu

## ABSTRACT

Neural ordinary differential equations (Neural ODEs) are an effective framework for learning dynamical systems from irregularly sampled time series data. These models provide a continuous-time latent representation of the underlying dynamical system where new observations at arbitrary time points can be used to update the latent representation of the dynamical system. Existing parameterizations for the dynamics functions of Neural ODEs limit the ability of the model to retain global information about the time series; specifically, a piece-wise integration of the latent process between observations can result in a loss of memory on the dynamic patterns of previously observed data points. We propose PolyODE, a Neural ODE that models the latent continuous-time process as a projection onto a basis of orthogonal polynomials. This formulation enforces long-range memory and preserves a global representation of the underlying dynamical system. Our construction is backed by favourable theoretical guarantees and in a series of experiments, we demonstrate that it outperforms previous works in the reconstruction of *past and future data*, and in downstream prediction tasks. Our code is available at https://github.com/edebrouwer/polyode.

## 1 INTRODUCTION

Time series are ubiquitous in many fields of science and as such, represent an important but challenging data modality for machine learning. Indeed, their temporal nature, along with the potentially high dimensionality makes them arduous to manipulate as mathematical objects. A long-standing line of research has thus focused on efforts in learning informative time series representations, such as simple vectors, that are capable of capturing local and global structure in such data (Franceschi et al., 2019; Gu et al., 2020). Such architectures include recurrent neural networks (Malhotra et al., 2017), temporal transformers (Zhou et al., 2021) and neural ordinary differential equations (neural ODEs) (Chen et al., 2018).

In particular, neural ODEs have emerged as a popular choice for time series modelling due to their sequential nature and their ability to handle irregularly sampled time-series data. By positing an underlying continuous time dynamic process, neural ODEs sequentially process irregularly sampled time series via piece-wise numerical integration of the dynamics between observations. The flexibility of this model family arises from the use of neural networks to parameterize the temporal derivative, and different choices of parameterizations lead to different properties. For instance, bounding the output of the neural networks can enforce Lipschitz constants over the temporal process (Onken et al., 2021).

The problem this work tackles is that the piece-wise integration of the latent process between observations can fail to retain a global representation of the time series. Specifically, each change to the hidden state of the dynamical system from a new observation can result in a loss of memory about prior dynamical states the model was originally in. This pathology limits the utility of neural ODEs when there is a necessity to retain information about the recent and distant past; *i.e.* current neural

ODE formulations are *amnesic*. We illustrate this effect in Figure 1, where we see that backward integration of a learned neural ODE (that is competent at forecasting) quickly diverges, indicating the state only retains sufficient local information about the future dynamics.

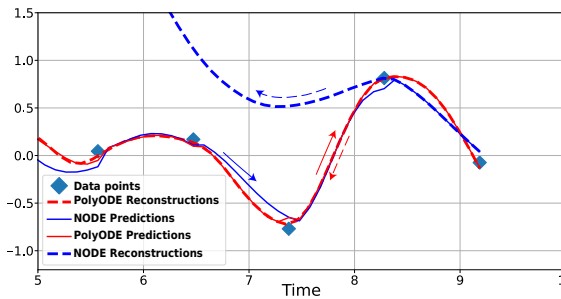

Figure 1: **PolyODE:** Illustration of the ability of PolyODE to reconstruct past trajectories. The solid lines show the forecasting trajectories conditioned on past observations for NODE (blue) and PolyODE (red). The dotted line represents the backward reconstruction for the past trajectories conditioned on the latent process at the last observation. We observe that PolyODE is able to accurately reconstruct the past trajectories while NODE quickly diverges. PolyODE is also more accurate in terms of forecasting.

One strategy that has been explored in the past to address this pathology is to regularize the model to encourage it to capture long-range patterns by reconstructing the time series from the last observation, using an auto-encoder architecture (Rubanova et al., 2019). This class of approaches results in higher complexity and does not provide any guarantees on the retention of the history of a time series. In contrast, our work proposes an alternative parameterization of the dynamics function that, *by design*, captures long-range memory within a neural ODE. Inspired by the recent successes of the HiPPO framework (Gu et al., 2020), we achieve this by enforcing that the dynamics of the hidden process follow the dynamics of the projection of the observed temporal process onto a basis of orthogonal polynomials. The resulting model, *PolyODE*, is a new neural ODE architecture that encodes long-range past information in the latent process and is thus *anamnesic*. As depicted in Figure 1, the resulting time series embeddings are able to reconstruct the past time series with significantly better accuracy.

**Contributions**   (1) We propose a novel dynamics function for a neural ODE resulting in PolyODE, a model that learns a global representation of high-dimensional time series and is capable of long-term forecasting and reconstruction by design. PolyODE is the first investigation of the potential of the HiPPO operator for neural ODEs architectures.

(2) Methodologically, we highlight the practical challenges in learning PolyODE and show how adaptive solvers for ODEs can overcome them. Theoretically, we provide bounds characterizing the quality of reconstruction of time series when using PolyODE.

(3) Empirically, we study the performance of our approach by assessing the ability of the learnt embeddings to reconstruct the past of the time series and by studying their utility as inputs for downstream predictive tasks. We show that our model provides better time series representations, relative to several existing neural ODEs architectures, based on the ability of the representations to accurately make predictions on several downstream tasks based on chaotic time series and irregularly sampled data from patients in intensive care unit.

## 2 RELATED WORK

*Time series modelling in machine learning:* There is vast literature on the use of machine learning for time series modelling and we highlight some of the ideas that have been explored to adapt diverse kinds of models for irregular time series data. Although not naturally well suited to learning representations of such data, there have been modifications proposed to discrete-time models such as recurrent neural networks (Hochreiter and Schmidhuber, 1997; Cho et al., 2014) to handle such data. Models such as mTANs (Shukla and Marlin, 2021) leverage an attention-based approach to interpolate sequences to create discrete-time data from irregularly sampled data. Another strategy has been architectural modifications to the recurrence equations e.g. CT-GRU (Mozer et al., 2017), GRU-D (Che et al., 2018) and Unitary RNNs (Arjovsky et al., 2016). Much more closely aligned to our work, and a natural fit for irregularly sampled data is research that uses differential equations to model continuous-time processes (Chen et al., 2018). By parameterizing the derivative of a time series using neural networks and integrating the dynamics over unobserved time points, this class of models is well suited to handle irregularly sampled data. This includes models such as ODE-RNN (Rubanova et al., 2019), ODE-LSTM (Lechner and Hasani, 2020) and Neural CDE (Kidger

et al., 2020). ODE-based approaches require the use of differential equation solvers during training and inference, which can come at the cost of runtime (Shukla and Marlin, 2021). PolyODEs lie in this family of models; specifically, this work proposes a new parameterization of the dynamics function and a practical method for learning that enables this model family to accurately forecast the future and reconstruct the past greatly enhancing the scope and utility of the learned embeddings.

*Orthogonal polynomials:* PolyODEs are inspired by a rich line of work in orthogonal decomposition of time series data. Orthogonal polynomials have been a mainstay in the toolkit for engineering (Heuberger et al., 2003) and uncertainty quantification (Li et al., 2011). In the context of machine learning, the limitations of RNNs to retain long-term memory have been studied empirically and theoretically (Zhao et al., 2020). Indeed, the GRU (Chung et al., 2014) and LSTM (Graves et al., 2007) architectures were created in part to improve the long-term memory of such models. Recent approaches for discrete-time models have used orthogonal polynomials and their ability to represent temporal processes in a memory-efficient manner. The Legendre Memory Unit (Voelker et al., 2019) and Fourier Recurrent Unit can be seen as a projection of data onto Legendre polynomials and Fourier basis respectively.

Our method builds upon and is inspired by the HiPPO framework which defines an operator to compute the coefficients of the projections on a basis of orthogonal polynomials. HiPPO-RNN and S4 are the most prominent examples of architectures building upon that framework (Gu et al., 2020; 2021). These models rely on a linear interpolation of the data in between observations, which can lead to a decrease of performance when the sampling rate of the input process is low. Furthermore, HiPPO-RNN and S4 perform the orthogonal polynomial projection of a non-invertible representation of the input data, which therefore doesn't enforce reconstruction in the observation space by design. Their design choices are motivated toward the goal of efficient mechanisms for capturing long term dependency for a target task (such as trajectory classification). In contrast, this work aims at exploring the abilities of the HiPPO operator for representation learning of irregular time series, when the downstream task is not known in advance.

Despite attempts to improve the *computational* performance of learning from long-term sequences (Morrill et al., 2021), to our knowledge, PolyODE is the first work that investigates the advantages of the HiPPO operator in the context of memory retention for continuous time architectures.

## 3 BACKGROUND

**Orthogonal Polynomial Projections:** Orthogonal polynomials are defined with respect to a measure $\mu$ as a sequence of polynomials $\{P_0(s), P_1(s), ...\}$ such that $deg(P_i) = i$ and

$$\langle P_n, P_m \rangle = \int P_n(s)P_m(s)d\mu(s) = \delta_{n=m}\alpha_n, \tag{1}$$

where $\alpha_n$ are normalizing scalars and $\delta$ is the Kronecker delta. For simplicity, we consider only absolutely continuous measures with respect to the Lebesgue measure, such that there exists a weight function $\omega(\cdot)$ such that $d\mu(s) = \omega(s)ds$. The measure $\mu$ determines the class of polynomials obtained from the conditions above (Eq. 1). Examples include Legendre, Hermite or Laguerre classes of orthogonal polynomials. The measure $\mu$ also defines an inner product $\langle \cdot, \cdot \rangle_\mu$ such that the orthogonal projection of a 1-dimensional continuous process $f(\cdot) : \mathbb{R} \to \mathbb{R}$ on the space of polynomials of degree $N$, $\mathcal{P}_N$, is given as

$$f_N(t) = \sum_{n=0}^{N} c_n P_n(t) \frac{1}{\alpha_n} \text{ with } c_n = \langle f, P_n \rangle_\mu = \int f(s)P_n(s)d\mu(s). \tag{2}$$

This projection minimizes the distance $\|f-p\|_\mu$ for all $p \in \mathcal{P}_N$ and is thus optimal with respect to the measure $\mu$. One can thus encode a process $f$ by storing its projection coefficients $\{c_0, ..., c_N\}$. We write the vector of coefficients up to degree $N$ as $\mathbf{c}$ (the degree $N$ is omitted) and $\mathbf{c}_i = c_i$. Intuitively, the measure assigns different weights at times of the process and thus allows for modulating the importance of different parts of the input signal for the reconstruction.

**Continuous update of approximation coefficients:** The projection of a process $f$ onto a basis of orthogonal polynomials provides an optimal representation for reconstruction. However, there is often a need to update this representation continuously as new observations of the process $f$ become available. Let $f_{<t}$ be the temporal process observed up until time $t$. We wish to compute

the coefficients of this process at different times $t$. We can define for this purpose a time-varying measure $\mu^t$ and corresponding weight function $\omega^t$ that can incorporate our requirements in terms of reconstruction abilities over time. For instance, if one cares about reconstruction of a process $\Delta$ temporal units in the past, one could use a time-varying weight function $\omega_t(s) = \mathbb{I}[s \in (t - \Delta, t)]$. This time-varying weight function induces a time-varying basis of orthogonal polynomials $P_n^t$ for $n = 0, ..., N$. We can define the time-varying orthogonal projection and its coefficients $c_n(t)$ as

$$f_{<t} \approx f_{<t,N} = \sum_{n=0}^{N} c_n(t) P_n^t \frac{1}{\alpha_n^t} \text{ with } c_n(t) = \langle f_{<t}, P_n^t \rangle_{\mu^t} = \int f_{<t}(s) P_n^t(s) d\mu^t(s). \quad (3)$$

**Dynamics of the projection coefficients:**   Computing the coefficients of the projection at each time step would be both computationally wasteful and would require storing the whole time series in memory, going against the principle of sequential updates to the model. Instead, we can leverage the fact that the coefficients evolve according to known linear dynamics over time. Remarkably, for a wide range of time-varying measures $\mu^t$, Gu et al. (2020) show that the coefficients $\mathbf{c}_N(t)$ follow:

$$\frac{dc_n(t)}{dt} = \frac{d}{dt} \int f_{<t}(s) P_n^t(s) d\mu^t(s), \quad \forall n \in \mathbb{N}$$

$$\frac{d\mathbf{c}(t)}{dt} = A_\mu \mathbf{c}(t) + B_\mu f(t) \quad (4)$$

where $A_\mu$ and $B_\mu$ are fixed matrices (for completeness, we provide a derivation of the relation for the translated Legendre measure in Appendix A). We use the translated Legendre measure in all our experiments. Using the dynamics of Eq. 4, it is possible to update the coefficients of the projection sequentially by only using the new incoming sample $f(t)$, while retaining the desired reconstruction abilities. Gu et al. (2020) use a discretization of the above dynamics to model discrete time-sequential data via a recurrent neural network architecture. Specifically, their architecture projects the hidden representation of an RNN onto a single time series that is projected onto an polynomial basis. Our approach differs in two ways. First, we work with a continuous time model. Second, we jointly model the evolution of $d$-dimensional time-varying process as a overparameterized hidden representation that uses orthogonal projections to serve as memory banks. The resulting model is a new neural ODE architecture as we detail below.

## 4 METHODOLOGY

**Problem Setup.** We consider a collection of sequences of temporal observations $\mathbf{x} = \{(\mathbf{x}_i, \mathbf{m}_i, t_i) : i \in \{1, ..., T\}\}$ that consist of a set of time-stamped observations and masks $(\mathbf{x}_i \in \mathbb{R}^d, \mathbf{m}_i \in \mathbb{R}^d, t_i \in \mathbb{R})$. We write $\mathbf{x}_{i,j}$ and $\mathbf{m}_{i,j}$ for the value of the $j^{\text{th}}$ dimension of $\mathbf{x}_i$ and $\mathbf{m}_i$ respectively. The mask $\mathbf{m}_i$ encodes the presence of each dimension at a specific time point. We set $\mathbf{m}_{i,j} = 1$ if $\mathbf{x}_{i,j}$ is observed and $\mathbf{m}_{i,j} = 0$ otherwise. The number of observations for each sequence $\mathbf{x}$, $T$, can vary across sequences. We define the set of sequences as $\mathcal{S}$ and the distance between two time series observed at the same times as $d(\mathbf{x}, \mathbf{x}') = \frac{1}{T} \sum_i^T \|\mathbf{x}_i - \mathbf{x}_i'\|_2$. Our goal is to be able to embed a sequence $\mathbf{x}$ into a vector $\mathbf{h} \in \mathbb{R}^{d_h}$ such that (1) $\mathbf{h}$ retains a maximal amount of information contained in $\mathbf{x}$ and (2) $\mathbf{h}$ is informative for downstream prediction tasks. We formalize both objectives below.

**Definition** (Reverse reconstruction). *Given an embedding $\mathbf{h}_t$ of a time series $\mathbf{x}$ at time $t$, we define the reverse reconstruction $\hat{\mathbf{x}}_{<t}$ as the predicted values of the time series at times prior to $t$. We write the observed time series prior to $t$ as $\mathbf{x}_{<t}$.*

**Objective 1** (Long memory representation). *Let $\mathbf{h}_t$ and $\mathbf{h}_t'$ be two embeddings of the same time series $\mathbf{x}$. Let $\hat{\mathbf{x}}_{<t}$ and $\hat{\mathbf{x}}_{<t}'$ be their reverse reconstruction. We say that $\mathbf{h}_t$ enjoys more memory than $\mathbf{h}_t'$ if $d(\hat{\mathbf{x}}_{<t}, \mathbf{x}_{<t}) < d(\hat{\mathbf{x}}_{<t}', \mathbf{x}_{<t})$.*

**Objective 2** (Downstream task performance). *Let $\mathbf{y} \in \mathbb{R}^{d_y}$ be an auxiliary vector drawn from a unknown distribution depending on $\mathbf{x}$. Let $\hat{\mathbf{y}}(\mathbf{x})$ and $\hat{\mathbf{y}}(\mathbf{x})'$ be the predictions obtained from embeddings $\mathbf{h}_t$ and $\mathbf{h}_t'$. For a performance metric $\alpha : \mathcal{S} \times \mathbb{R}^{d_y} \to \mathbb{R}$, we say that $\mathbf{h}_t$ is more informative than $\mathbf{h}_t'$ if $\mathbb{E}_{\mathbf{x},\mathbf{y}}[\alpha(\hat{\mathbf{y}}(\mathbf{x}), \mathbf{y})] > \mathbb{E}_{\mathbf{x},\mathbf{y}}[\alpha(\hat{\mathbf{y}}(\mathbf{x})', \mathbf{y})]$.*

### 4.1 POLYODE: ANAMNESIC NEURAL ODES

We make the assumption that the observed time series $\mathbf{x}$ comes from an unknown but continuous temporal process $\mathbf{x}(t)$. Given $\mathbf{h}(t) \in \mathbb{R}^{d_h}$ and a read-out function $g : \mathbb{R}^{d_h} \to \mathbb{R}^d$ we posit the

following generative process for the data:

$$\mathbf{x}(t) = g(\mathbf{h}(t)), \qquad \frac{d\mathbf{h}(t)}{dt} = \phi(\mathbf{h}(t)) \tag{5}$$

where part of $\phi(\cdot)$ is parametrized via a neural network $\phi_\theta(\cdot)$.

The augmentation of the state space is a known technique to improve the expressivity of Neural ODEs (Dupont et al., 2019). Here, to ensure that the hidden representation in our model has the capacity to retain long-term memory, we augment the state space of our model by including the dynamics of coefficients of orthogonal polynomials as described in Equation 4.

Similarly as classical filtering architectures (e.g. Kalman filters and ODE-RNN (Rubanova et al., 2019)), PolyODE alternates between two regimes : an integration step (that takes place in between observations) and an update step (that takes place at the times of observations), described below.

We structure the hidden state as $\mathbf{h}(t) = [\mathbf{h}_0(t), \mathbf{h}_1(t), \ldots, \mathbf{h}_d(t)]$ where $\mathbf{h}_0(t) \in \mathbb{R}^d$ has the same dimension as the input process $\mathbf{x}$, $\mathbf{h}_i(t) \in \mathbb{R}^N, \forall i \in 1, \ldots, d$, has the same dimension as the vector of projection coefficients $\mathbf{c}^i(t)$ and $[\cdot, \cdot]$ is the concatenation operator. We define the readout function $g_i(\cdot) : \mathbb{R}^{(N+1)d} \to \mathbb{R}$ such that $g_i(\mathbf{h}(t)) = \mathbf{h}_0(t)_i$. That is, $g_i$ is fixed and returns the $i^{\text{th}}$ value of the input vector. This leads to the following system of ODEs that characterize the evolution of $\mathbf{h}(t)$:

**Integration Step.**

$$\begin{cases} \frac{d\mathbf{c}^1(t)}{dt} &= A_\mu \mathbf{c}^1(t) + B_\mu g_1(\mathbf{h}(t)) \\ \qquad \vdots \\ \frac{d\mathbf{c}^d(t)}{dt} &= A_\mu \mathbf{c}^d(t) + B_\mu g_d(\mathbf{h}(t)) \\ \frac{d\mathbf{h}(t)}{dt} &= \phi_\theta(\mathbf{h(t)}) \end{cases} \tag{6}$$

This parametrization allows learning arbitrarily complex dynamics for the temporal process $\mathbf{x}$. We define a sub-system of equations of projection coefficients update for each dimension of the input temporal process $\mathbf{x}(t) \in \mathbb{R}^d$. This sub-system is equivalent to Equation 4, where we have substituted the input process by the prediction from the hidden process $\mathbf{h}(t)$ through a mapping $g_i(\cdot)$. The hidden process $\mathbf{h}_0(t)$ acts similarly as in a classical Neural ODEs and the processes $\mathbf{c}(t)$ captures long-range information about the observed time series. During the integration step, we integrate both the hidden process $\mathbf{h}(t)$ and the coefficients $\mathbf{c}(t)$ forward in time, using the system of Equation 6. At each time step, we can provide an estimate of the time series $\hat{\mathbf{x}}(t)$ conditioned on the hidden process $\mathbf{h}(t)$, with $\hat{\mathbf{x}}(t) = g(\mathbf{h}(t))$.

The coefficients $\mathbf{c}(t)$ are influenced by the values of $\mathbf{h}(t)$ through $\mathbf{h}_0(t)$ only. The process $\mathbf{h}_0(t)$ provides the signal that will be memorized by projecting onto the orthogonal polynomial basis. The $\mathbf{c}(t)$ serve as memory banks and do not influence the dynamics of $\mathbf{h}(t)$ during the integration step.

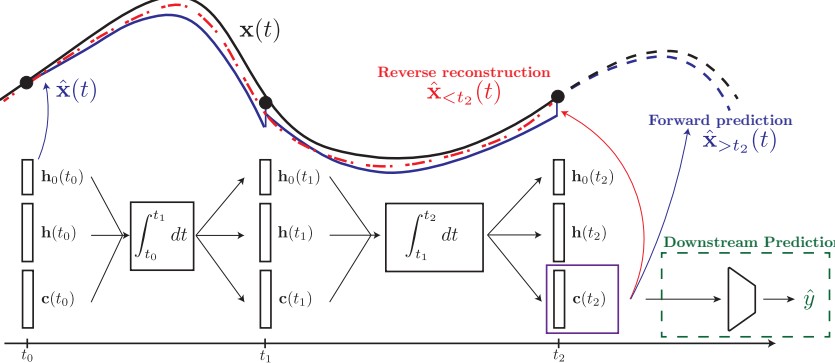

Figure 2: **PolyODE time series embedding process**. The model processes the time series sequentially by alternating between integration steps (between observations) and update steps when observations are collected. Informative embeddings should allow for (1) reconstructing the past of the time series (reverse reconstruction - in red), (2) forecasting the future of the sequence (forward prediction - in blue) and (3) being informative for downstream predictions (in green).

The system of equations in Eq. 6 characterises the dynamics in between observations. When a new observation becomes available, we update the system as follows.

**Update Step.** *At time $t = t_i$, after observing $\mathbf{x}_i$ and mask $\mathbf{m}_i$, we set*

$$\begin{cases} \mathbf{h}_j(t_i) := \mathbf{c}^j(t_i), & \forall j \; s.t. \; \mathbf{m}_{i,j} = 1 \\ \mathbf{h}_0(t_i)_j := \mathbf{x}_{i,j}, & \forall j \; s.t. \; \mathbf{m}_{i,j} = 1 \end{cases} \tag{7}$$

The update step serves the role of incorporating new observations in the hidden representation of the system. It proceeds by (1) reinitializing the hidden states of the system with the orthogonal polynomial projection coefficients $\mathbf{c}(t)$: $\mathbf{h}_j(t_i) := \mathbf{c}^j(t_i)$; and (2) resetting $\mathbf{h}_0(t)$ to the newly collected observation: $\mathbf{h}_0(t_i)_j := \mathbf{x}_{i,j}(t)$.

**Remarks:** Our model blends orthogonal polynomials with the flexibility offered in modelling the observations with NeuralODEs. The consequence of this is that while the coefficients serve as memory banks for each dimension of the time series, the Neural ODE over $\mathbf{h}_0(t)$ can be used to forecast from the model. That said, we acknowledge that a significant limitation of our current design is the need for the hidden dimension to track $N$ coefficients for each time-series dimension. Given that many adjacent time series might be correlated, we anticipate that methods to reduce the space footprint of the coefficients within our model is fertile ground for future work.

## 4.2 TRAINING

We train this architecture by minimizing the reconstruction error between the predictions and the observations: $\mathcal{L} = \sum_{i=1}^{T} \|\hat{\mathbf{x}}(t_i) - \mathbf{x}_i\|_2^2$. We first initialize the hidden processes $\mathbf{c}(0) = 0$ and $\mathbf{h}(0) = 0$ though they can be initialized with static information $b$, if available (*e.g.* $\mathbf{h}(0) = \psi_\theta(b)$). We subsequently alternate between integration steps between observations and update steps at observation times. The loss is updated at each observation time $t_i$. A pseudo-code description of the overall procedure is given in Algorithm 1.

**Numerical integration.** We integrate the system of differential equations of Equation 6 using differentiable numerical solvers as introduced in Chen et al. (2018). However, one of the technical challenges that arise with learning PolyODE is that the dynamical system in Equation 6 is relatively stiff and integrating this process with acceptable precision would lead to prohibitive computation times with explicit solvers. To deal with this instability we used an implicit solver such as Backward Euler or Adams-Moulton for the numerical integration (Sauer, 2011). A comparison of numerical integration schemes and an analysis of the stability of the ODE are available in Appendix I.

---

**Algorithm 1:** PolyODE Training

**Data:** $\mathbf{x}$, matrices $A_\mu, B_\mu$, number of dimensions $d$, number of observations $T$, number of polynomial coefficients $N$

**Result:** Training loss $\mathcal{L}$ over a whole sequence $\mathbf{x}$

$t^* \leftarrow 0$

Initialize $\mathbf{h}_j(0) = \mathbf{c}_j(0) = \mathbf{0}_N, \forall j \in 1, ..., d$,

Loss $\mathcal{L} = 0$

**for** $i \leftarrow 1$ **to** $T$ **do**

    **Integrate** $\mathbf{c}_{1,...,d}(t)$ and $\mathbf{h}_{0,...,d}(t)$ from $t = t^*$ until $t = t_i$

    $\hat{\mathbf{x}}_i \leftarrow \mathbf{h}_0(t^*)$

    **Update** $\mathbf{c}_{1,...,d}(t_i)$ and $\mathbf{h}_{0,...,d}(t_i)$ with $\mathbf{x}_i, \mathbf{m}_i$.

    $\mathcal{L} = \mathcal{L} + \|(\hat{\mathbf{x}}_i - \mathbf{x}_i) \odot \mathbf{m}_i\|_2^2$

    $t^* \leftarrow t_i$

**end**

---

**Forecasting:** From time $t$, we forecast the time series at an arbitrary time $t^*$ as:

$$\hat{\mathbf{x}}_{>t}(t^*) = g(\mathbf{h}(t) + \int_t^{t^*} \phi_\theta(\mathbf{h}(s))ds), \tag{8}$$

where $\phi_\theta(\cdot)$ is the learned model that we use in the integration step and introduced in Eq. 5.

**Reverse Reconstruction:** Using Equation 3, we can compute the reverse reconstruction of the time series at any time $t$ using the projection coefficients part of the hidden process:

$$\hat{\mathbf{x}}_{<t,j} = \sum_{n=0}^{N} c_n^j(t) \cdot P_n^t \cdot \frac{1}{\alpha_n^t}. \tag{9}$$

More details about this reconstruction process and its difference with respect to classical NODEs are available in Appendix E. The error between the prediction obtained during the integration step, $\hat{\mathbf{x}}(t)$, and the above reconstruction estimator is bounded above, as Result 4.1 shows.

**Result 4.1.** *For a shifted rectangular weighting function with width $\Delta$, $\omega^t(x) = \frac{1}{\Delta}\mathbb{I}_{[t-\Delta,t]}$ (which generate Legendre polynomials), the mean square error between the forward ($\hat{\mathbf{x}}$) and reverse prediction ($\hat{\mathbf{x}}_{<t}$) at each time $t$ is bounded by:*

$$\|\hat{\mathbf{x}} - \hat{\mathbf{x}}_{<t}\|_{\mu^t}^2 \leq C_0 \frac{\Delta^2 L^2 (K+1)^2}{N(2N-1)} + C_1 \Delta L(K+1) S_K \xi\left(\frac{3}{2}, N\right) + C_2 S_K^2 \xi\left(\frac{3}{2}, N\right),$$

*where $K$ is the number of observations in the interval $[t - \Delta, t]$, $L$ is the Lipschitz constant of the forward process, $N$ is the degree of the polynomial approximation and $\xi(\cdot, \cdot)$ is the Hurwitz zeta function. $S_K = \sum_{i=1}^{K}|\hat{\mathbf{x}} - \mathbf{x_i}|$ is the sum of absolute errors between the forward process and observations incurred at the update steps. $C_0, C_1$ and $C_2$ are constants.*

Expectedly, the bound goes to $0$ as the degree of the approximation increases. The lower cumulative absolute error $S_K$ also leads to a reduction of this bound. As the cumulative absolute error $S_K$ and our loss function $\mathcal{L}$ share the same optimum, for fixed $\Delta$, $L$, $K$ and $N$, our training objective therefore implicitly enforces a minimization of the reconstruction error. This corresponds to optimizing Objective 1, where we set $d(\mathbf{x}, \mathbf{x}') = \|\mathbf{x} - \mathbf{x}'\|_{\mu^t}^2$. Our architecture thus jointly minimizes both global reconstruction and forecasting error. Notably, when $S_K = 0$, this result boils down to the well-known projection error for orthogonal polynomials projection of continuous processes (Canuto and Quarteroni, 1982). What is more, increasing the width of the weighting function (increasing $\Delta$) predictably results in higher reconstruction error. However, this can be compensated by increasing the dimension of the polynomial basis accordingly. We also note a quadratic dependency on the Lipschitz constant of the temporal process, which can limit the reverse reconstruction abilities for high-frequency components. The full proof can be found in Appendix B.

## 5 EXPERIMENTS

We evaluate our approach on two objectives : (1) the ability of the learned embedding to encode global information about the time series, through the reverse reconstruction performance (or memorization) and (2) the ability of embedding to provide an informative input for a downstream task. We study our methods on the following datasets:

**Synthetic Univariate**. We validate our approach using a univariate synthetic time series. We simulate 1000 realizations from this process and sample it at irregularly spaced time points using a Poisson point process. For each generated irregularly sampled time series $\mathbf{x}$, we create a binary label $y = \mathbb{I}[x(5) > 0.5]$. Further details about datasets are to be found in Appendix G.

**Chaotic Attractors**. Chaotic dynamical systems exhibit a large dependence of the dynamics on the initial conditions. This means that a noisy or incomplete evaluation of the state space may not contain much information about the past of the time series. We consider two widely used chaotic dynamical systems: Lorenz63 and a 5-dimensional Lorenz96. We generate 1000 irregularly sampled time series from different initial conditions. We completely remove one dimension of the time series such that the state space is never fully observed. This forces the model to remember the past trajectories to create an accurate estimate of the state space at each time $t$.

**MIMIC-IIII dataset**. We use a pre-processed version of the MIMIC-III dataset (Johnson et al., 2016; Wang et al., 2020). This consists of the first 24 hours of follow-up for ICU patients. For each time series, the label $y$ is the in-hospital mortality.

*Baselines:* We compare our approach against two sets of baselines: Neural ODEs architecture and variants of recurrent neural networks architectures designed for long-term memory. To ensure a fair comparison, we use the same dimensionality of the hidden state for all models.

**Neural ODE baselines**. We use a filtering implementation of Neural ODEs, *GRU-ODE-Bayes* (De Brouwer et al., 2019) and *ODE-RNN* (Rubanova et al., 2019), an auto-encoder relying on a Neural ODE for both the encoder and the decoder part. For theses baselines, we compute the reverse reconstruction by integrating the system of learnt ODEs backward in time. In case of ODE-RNN, we use the ODE of the decoder. Additionally, we compare against Neural RDE neural controlled differential equations for long time series (Neural RDE) (Morrill et al., 2021).

**Long-term memory RNN baselines**. We compare against *HiPPO-RNN* (Gu et al., 2020), a recurrent neural network architecture that uses orthogonal polynomial projections of the hidden process. We also use a variant of this approach where we directly use the HiPPO operator on the observed time series, rather than on the hidden process. We call this variant *HiPPO-obs*. We also compare against S4, an efficient state space model relying on the HiPPO matrix (Gu et al., 2021).

*Long-range representation learning:* For each dataset, we evaluate our method and the various baselines on different tasks. Implementation details are available in Appendix H.

**Downstream Classification.** We train the models on the available time series. After training, we extract time series embedding from each model and use them as input to a multi-layer perceptron trained to predict the time series label $y$. We report the area under the operator-characteristic curve evaluated on a left-out test set with 5 repetitions.

**Time Series Reconstruction.** Similarly as for the downstream classification, we extract the time series embeddings from models trained on the time series. We then compute the reverse reconstruction $\hat{\mathbf{x}}_{<t}$ and evaluate the MSE with respect to the true time series.

**Forecasting.** We compare the ability of all models to forecast the future of the time series. We compute the embedding of the time series observed until some time $t_{\text{cond}}$ and predict over a horizon $t_{\text{horizon}}$. We then report the MSE between the prediction and true trajectories.

Table 1: Downstream task and reverse reconstruction results for synthetic and Lorenz datasets.

| Model | Downstream Classification↑ | | | Reconstruction↓ | | |
|---|---|---|---|---|---|---|
| | Synthetic | Lorenz63 | Lorenz96 | Synthetic | Lorenz63 | Lorenz96 |
| Irregular Rate $\lambda$ | 0.7 | 0.3 | 0.3 | 0.7 | 0.3 | 0.3 |
| GRU-ODE | $0.968 \pm 0.004$ | $0.825 \pm 0.031$ | $0.925 \pm 0.004$ | $0.057 \pm 0.010$ | $0.752 \pm 0.057$ | $0.346 \pm 0.072$ |
| ODE-RNN | $0.870 \pm 0.032$ | $0.813 \pm 0.013$ | $0.954 \pm 0.012$ | $0.080 \pm 0.036$ | $0.674 \pm 0.049$ | $0.214 \pm 0.030$ |
| Neural-RDE | $0.773 \pm 0.111$ | $0.604 \pm 0.046$ | $0.606 \pm 0.112$ | $0.167 \pm 0.031$ | $0.989 \pm 0.074$ | $1.747 \pm 0.472$ |
| HiPPO-obs | $0.758 \pm 0.023$ | $0.837 \pm 0.034$ | $0.949 \pm 0.007$ | $0.197 \pm 0.010$ | $0.511 \pm 0.043$ | $0.247 \pm 0.005$ |
| HiPPO-RNN | $0.742 \pm 0.008$ | $0.804 \pm 0.023$ | $0.944 \pm 0.008$ | $0.209 \pm 0.018$ | $0.784 \pm 0.122$ | $0.198 \pm 0.014$ |
| S4 | $\mathbf{0.994 \pm 0.003}$ | $0.911 \pm 0.005$ | $0.948 \pm 0.016$ | $0.032 \pm 0.006$ | $0.428 \pm 0.040$ | $0.171 \pm 0.008$ |
| **PolyODE** | $\mathbf{0.994 \pm 0.003}$ | $\mathbf{0.992 \pm 0.000}$ | $\mathbf{0.984 \pm 0.002}$ | $\mathbf{0.012 \pm 0.002}$ | $\mathbf{0.034 \pm 0.008}$ | $\mathbf{0.038 \pm 0.008}$ |

Results for these tasks are presented in Table 1 for Synthetic and Lorenz datasets and in Table 2 for MIMIC. We report additional results in Appendix C, with a larger array of irregular sampling rates. We observe that the reconstruction abilities of PolyODE clearly outperforms the other baselines, for all datasets under consideration. A similar trend is to be noted for the downstream classification for the synthetic and Lorenz datasets. For these datasets, accurate prediction of the label $y$ requires a global representation of the time series, which results in better performance for our approach.

For the MIMIC dataset, our approach compares favourably with the other methods for the downstream classification objective and outperforms other methods for trajectory forecasting. What is more, the reconstruction ability of PolyODE is significantly better than compared approaches. In Figure 3, we plot the reverse

Table 2: Performance on MIMIC-III dataset.

| Method | Classification ↑ | Forecasting ↓ | Reconstruction ↓ |
|---|---|---|---|
| HiPPO-obs | $0.793 \pm 0.002$ | / | $0.775 \pm 0.000$ |
| HiPPO-RNN | $0.764 \pm 0.006$ | $1.104 \pm 0.009$ | $0.969 \pm 0.026$ |
| GRU-ODE | $0.793 \pm 0.005$ | $1.413 \pm 0.074$ | $2025.6 \pm 2365.1$ |
| ODE-RNN | $\mathbf{0.800 \pm 0.004}$ | $1.104 \pm 0.026$ | $6.343 \pm 4.844$ |
| **PolyODE** | $0.778 \pm 0.005$ | $\mathbf{1.085 \pm 0.022}$ | $\mathbf{0.187 \pm 0.005}$ |

reconstructions of PolyODE for several vitals of a random patient over the first 24 hours in the ICU. This reconstruction is obtained by first sequentially processing the time series until $t = 24$ hours and subsequently using the hidden process to reconstruct the time series as in Equation 9. We observe that PolyODE can indeed capture the overall trend of the time series over the whole history.

*Ablation study - the importance of the auxiliary dynamical system:* Is there utility in leveraging the neural network $\phi_\theta(\cdot)$ to learn the dynamics of the time series? How well would various interpolation

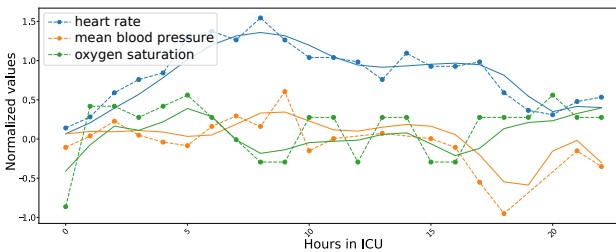

Figure 3: **PolyODE**: Reverse prediction of vitals over the 24 hours of ICU of a randomly selected test patient. We plot the true value (dots) and reconstructions (solid line) for different vitals. Reverse reconstruction is done from the last time observation. Other vitals are provided in Appendix D.

schemes for irregularly sampled observations perform in the context of reverse reconstruction and classification? In response to these questions, we first note that they do not support extrapolation and are thus incapable of forecasting the future of the time series. However, we compare the performance in terms of reverse reconstruction and classification in Table 3. We consider constant interpolation (last observation carried forward), linear interpolation and Hermite spline interpolation. Our results indicate a significant gap in performance between PolyODE and the linear and constant interpolation schemes. The Hermite spline interpolation allows us to capture most of the signal needed for the downstream classification task but results in significantly lower performance in terms of the reverse reconstruction error. These results therefore strongly support the importance of $\phi_\theta(\cdot)$ for producing informative time series embeddings. Complementary results are available in Appendix C.

Table 3: Impact of the interpolation scheme on performance.

| | Downstream Classification↑ | | | Reconstruction↓ | | |
|---|---|---|---|---|---|---|
| | SimpleTraj | Lorenz | Lorenz96 | SimpleTraj | Lorenz | Lorenz96 |
| Irregular Rate $\lambda$ | 0.7 | 0.3 | 0.3 | 0.7 | 0.3 | 0.3 |
| Constant | $0.969 \pm 0.005$ | $0.664 \pm 0.033$ | $0.862 \pm 0.017$ | $0.027 \pm 0.003$ | $0.785 \pm 0.074$ | $0.393 \pm 0.017$ |
| Linear | $0.969 \pm 0.008$ | $0.744 \pm 0.016$ | $0.857 \pm 0.026$ | $0.028 \pm 0.005$ | $0.787 \pm 0.066$ | $0.388 \pm 0.032$ |
| Hermite Spline | $0.971 \pm 0.012$ | $0.976 \pm 0.000$ | $\mathbf{0.983 \pm 0.004}$ | $0.055 \pm 0.016$ | $0.135 \pm 0.007$ | $0.093 \pm 0.011$ |
| **PolyODE** | $\mathbf{0.994 \pm 0.003}$ | $\mathbf{0.992 \pm 0.000}$ | $\mathbf{0.984 \pm 0.002}$ | $\mathbf{0.012 \pm 0.002}$ | $\mathbf{0.034 \pm 0.008}$ | $\mathbf{0.038 \pm 0.008}$ |

*Incorporating global time series uncertainty:* Previous experiments demonstrate the ability of PolyODE to retain memory of the past trajectory. A similar capability can be obtained for capturing global model uncertainties over the time series history. In Figure 4, we evaluate the association between the recovered uncertainties of PolyODE and the reverse reconstruction errors. We plot the predicted uncertainties against the root mean square error (RMSE) on a logarithmic scale. We compare our approach with using the uncertainty of the model at the last time step only. We observe that the uncertainties recovered by PolyODE are significantly more correlated with the errors (Pearson-$\rho = 0.56$) compared to using the uncertainties obtained from the last time step (Pearson-$\rho = 0.11$). More details are available in Appendix F.

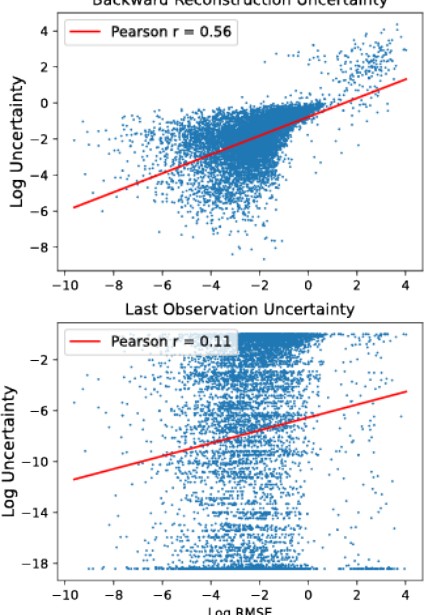

Figure 4: Association between uncertainties and reverse reconstruction errors for PolyODE (top) and classical Neural ODEs (bottom).

## 6 CONCLUSION

Producing time series representations that are easy to manipulate, representative of global dynamics, practically useful for downstream tasks and robust to irregular sampling remains an ongoing challenge. In this work, we took a step in that direction by proposing a simple but novel architecture that satisfies those requirements by design. As a Neural ODE, PolyODE inherits the ability to handle irregular time series elegantly but at the same time, PolyODE also incurs computational cost associated with numerical integration. Currently, our approach also requires a large hidden space dimension and finding methods to address this that exploit the correlation between dimensions of the time series is a fruitful direction for future work.

**Reproducibility Statement** Details for reproducing experiments shown are available in Appendix H. The code for reproducing all experiments will be made publicly available.

**Acknowledgements**

EDB is funded by a FWO-SB PhD research grant (S98819N) and a FWO research mobility grant (V424722N). RGK was supported by a CIFAR AI Chair. Resources used in preparing this research were provided, in part, by the Province of Ontario, the Government of Canada through CIFAR, and companies sponsoring the Vector Institute.

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
