# OpenReview forum: "Anamnesic Neural Differential Equations with Orthogonal Polynomial Projections"
_ICLR.cc/2023/Conference — ICLR 2023 poster_

### Official Review · Reviewer_uK5k · 2022-10-25

**Confidence:** 5
**Correctness:** 3
**Technical Novelty And Significance:** 2
**Empirical Novelty And Significance:** 2
**Recommendation:** 6

**Clarity, Quality, Novelty And Reproducibility:**

Post rebuttal update:
---------------------

Quality/Novelty: While largely building off prior work, this paper does propose a new method that combines two established techniques from the literature (HiPPO and neural ODEs). I think the method itself is an interesting proposition.

Clarity: I am an expert on the HiPPO line of work and have a general familiarity with neural ODEs, yet I found the technical details of the paper and proposed method incredibly hard to understand, which were only somewhat clarified by a detailed back-and-forth with the authors. I think that the description and comparison to prior work is also somewhat mischaracterized, e.g. claiming that prior work involves memorizing a latent state instead of the input signal, which is a conflation of two different things (HiPPO vs HiPPO-RNN). Part of this confusing description arises from a lack of transparency about the tradeoffs of the method, in particular computation speed. Overall, I think that clarity of both the method and its relation to prior work is the major weakness of this paper.

Reproducibility: Despite the lengthy exchange in this thread, I am not confident enough in several details to be able to implement the method from scratch. However, supplementary code was provided and I believe that the authors intend to release reproducible experiments.

--------------

Final post-rebuttal update:

The authors have engaged in a lengthy discussion that have cleared up details of the paper, and taken significant efforts to improve the presentation of the paper and the comparisons to related work. I have increased my recommendation to acceptance.


**Strength And Weaknesses:**

This paper is not well described or positioned. It appears to be the exactly same as the HiPPO (High Order Polynomial Projections) method from the literature, with two minor variations:
1. It considers $d$-dimensional input instead of $1$-dimensional, by simply applying $d$ independent HiPPO models.
2. It claims to be "continuous time" and uses a different numerical integration step in between observations, although here the technical details are missing so it is not clear what precisely is being proposed.

Several technical details are missing from the paper.
- It is not clear what $A_\mu$ means in the paper. It is mentioned that HiPPO derives several families of matrices based on a measure $\mu$; which one is used in this paper? Is it initialized to a specific matrix and then learned, or frozen to a HiPPO matrix?
- Details around the "Numerical Integration" are very confusing and not clearly presented anywhere in the paper. Section 4.2 claims to use a numerical solver in the style of Neural ODEs, but later claims to use a solver such as Backward Euler. If it is the latter, then Backward Euler is a special case of the integration rules considered in the original HiPPO paper. If it is the former, a more detailed comparison of the tradeoffs of the different integration rules should be discussed and ablated (in particular, a black-box ODE solver trained with the adjoint method should be much slower than the linear recurrences used in HiPPO).
- How are reconstructions for NODE being done in Figure 1? The PolyODE appears to be just the HiPPO reconstruction framework, but such closed form reconstruction formulas don't exist for general NODEs. Also, what is \alpha in equation (9)?

Even if the method was described more clearly, this application of HiPPO to higher-dimensional data raises more points that should be discussed. The paper acknowledges that a major limitation of this method is the computational expense of using $d$ independent features (requiring a state size blown up to $dN$), which is why the original HiPPO only used dimension $1$.
- Computational tradeoffs of this method and baselines should be discussed and quantitatively compared, particularly as a function of $d$ and $N$. Additionally, the numerical integration method should be more clear and its computational speed ablated, as it seems to be an interchangeable component compared to HiPPO's discretization method.
- This issue of blowing up the dimension by a factor of $N$ is in fact one of the main problems addressed by later extensions of HiPPO to full state-space models such as S4. These baselines should be included where appropriate, as they can be used as drop-in models for most of the considered applications (note that they can also be applied to many "irregularly-sampled" time series problems where the sampling is provided as a mask).

**Summary Of The Paper:**

This paper combines techniques from Neural ODEs together with with prior work on polynomial projections for signal reconstruction (HiPPO). It defines a new method PolyODE that can address various problems with time series analysis such as reconstruction and forecasting, particularly for settings such as irregularly-sampled data.




**Summary Of The Review:**

This paper contributes technical ideas that improve on baselines on addressing the problems it is concerned about (e.g. irregular sampled time series). However, the method description is incredibly confusing, which also partly stems from inaccurate portrayals of prior work.

---

> ### Author Response · Authors · 2022-11-18
> **Comment about the positioning and contributions of our work**
>
> We want to thank reviewer uK5k for his comments. We are glad to report that we have implemented their suggestions in the updated version of the manuscript. We hope the following answers to your comments will help answer the concerns raised in the review.
>
> First, we fully acknowledge our reliance on the HiPPO framework. Indeed, as reviewer msc4 notes, our work “extends from Gu et al. 2020 into neural ODE”.
> We are sorry to read that this does not appear as clearly as we thought in the initial submission.
> We have updated the introduction and related works sections to clarify this dependency.
> That said, while we build upon this framework, our work significantly differs from the previous HiPPO papers. Indeed, our work makes the following contributions:
>
> - Contribution to continuous time modeling: We propose a *Neural ODE* architecture, and therefore extends upon existing Neural ODE works. Importantly, it addresses important limitations of previous popular neural ODE models, the fact that existing approaches are *amnesic*: each change to the hidden state of the dynamical system from a new observation can result in a loss of memory about prior dynamical states the model was originally in. This motivation is graphically depicted in Figure 1. Our framework therefore leverages the successes of the HiPPO literature (instantiated in discrete time) for improving Neural ODE architectures for continuous time data while provided theoretical guarantees on the associated reconstruction error.
>
> - Rationale for continuous time modeling: To motivate why the extension of the ideas in HIPPO to continuous time data is vital we highlight that the HiPPO and S4 models use a recurrent architecture. Our Neural ODE framework allows us to operate with arbitrary sampling intervals by learning how the dynamics of the time series in between observations. The dynamics are learnt by leveraging the availability of other realizations of time series. The HiPPO and the S4 models take a different route by assuming a linear interpolation in between observations. While this strategy can accommodate irregular sampling with short intervals, our empirical results show that it fails when the sampling rate is low.
>
> - Contribution to representation learning: In contrast to the HiPPO / S4 framework which aims at processing long time series for specific tasks (such as classification or regression), we focus on the problem of time series *representation learning*. As stated at the beginning of our methodology section, we want models that provide both long memory representation and downstream task performance. Our goal is to learn meaningful embeddings of time series that can be used for other downstream tasks, as highlighted in our experiments. Our models are trained to forecast the future of the time series. The resulting embeddings are then frozen and used for downstream predictions. As such, our model are trained in an *unsupervised* way.  In contrast, the HiPPO and S4 framework are designed to capture long range dependencies in *supervised* settings. Unless trained to do so, these models do not necessarily produce embedding that can capture the whole past of the trajectory. Indeed, the HiPPO and S4 framework perform the orthogonal polynomial in the *latent space* while we perform it on the *observation space*. This crucial difference leads to reconstruction guarantees of the time series, as our Result 4.1 shows. Because one cannot invert the latent vectors back to the observation space, HiPPO and S4 fail to provide accurate backward reconstruction.

---

> > ### Author Response · Authors · 2022-11-18
> > **Answers to reviewer's comments (1/2)**
> >
> > ### About matrix $A_{\mu}$
> >
> > `It is not clear what $A_{\mu}$ means in the paper. It is mentioned that HiPPO derives several families of matrices based on a measure μ; which one is used in this paper? Is it initialized to a specific matrix and then learned, or frozen to a HiPPO matrix?`
> >
> > We use the translated legendre measure in our experiments. We clarified this in the last paragraph of Section 3 in the paper. The width of the weight function for each experiment, along with the other hyper-parameters used is also given in Table 7 of Appendix G.
> >
> > ### About numerical integration
> >
> > `Details around the "Numerical Integration" are very confusing and not clearly presented anywhere in the paper. Section 4.2 claims to use a numerical solver in the style of Neural ODEs, but later claims to use a solver such as Backward Euler. If it is the latter, then Backward Euler is a special case of the integration rules considered in the original HiPPO paper. If it is the former, a more detailed comparison of the tradeoffs of the different integration rules should be discussed and ablated (in particular, a black-box ODE solver trained with the adjoint method should be much slower than the linear recurrences used in HiPPO).`
> >
> >
> > Thank you for this comment. We have now added a brief description of several numerical integration methods in Appendix H where  we described the Euler method, the Dopri-5 method and the Adams-Moulton method.
> >
> > To answer your question directly, our experiments used Adams-Moulton of order 4. Backward Euler and the trapezoidal rule (as used in the HiPPO paper) are indeed special cases of Adams-Moulton (order 0 and 1). Our contribution does not lie in choosing an order 4 over an order 1 integrator method but rather in providing a way to continuously interpolate in between the observations (as stated above).
> >
> > To complement our experiments, we have performed a comparison of these solvers both in terms of computation time and in terms of performance (validation loss on the forecasting task). We show that the Euler method is unstable and leads to very large integration errors. On the other hand, the Dopri-5 method, that can adjust the step size to keep the error under control, is prohibitively expensive (an order of magnitude slower than Euler and Adams-Moulton). Finally, we also added details about the stability of the system of Neural ODE, which explains these experimental findings. We report the spectral signature of the matrix $A_{\mu}$ and show that its stiffness ratio grows with the number of projection coefficients $N$.
> >
> > ### About the reconstruction process in PolyODEs
> > `How are reconstructions for NODE being done in Figure 1? The PolyODE appears to be just the HiPPO reconstruction framework, but such closed form reconstruction formulas don't exist for general NODEs. Also, what is \alpha in equation (9)?`
> >
> > The reconstructions of Figure 1 use the orthogonal polynomial reconstruction. The equation for the reverse reconstruction is given in Equation 9. At the last time step of Figure 9 (t=9.2), we use the orthogonal polynomial projection coefficients $c_n(t=9.2)$ to produce :
> > $\hat{\mathbf{x}}_{<t} = \sum_{n=0}^N c_n(t) \cdot P_n^t \cdot \frac{1}{\alpha_n^t}.$
> >
> > In Equation 9, alpha refers to the scaling of the basis of orthogonal polynomials, as introduced in Equation 1. Canonical orthogonal polynomials are not normalized (e.g. $\alpha_n = 2 / (2n+1)$ for Legendre polynomials).
> >
> > ### About tracking each of the $d$ temporal dimensions
> > `Even if the method was described more clearly, this application of HiPPO to higher-dimensional data raises more points that should be discussed. The paper acknowledges that a major limitation of this method is the computational expense of using d independent features (requiring a state size blown up to dN), which is why the original HiPPO only used dimension 1`
> >
> > As we acknowledge in the paper, the need for tracking the $d$ time series features leads to a significant increase in the dimension of the hidden state. Yet, this is necessary if we want to enforce long memorization of the time series by design. While the HiPPO paper circumvents this difficulty by projecting the hidden space to a single dimensional process, it also loses the inherent property of long-term memorization. To be clear, long-term memory can be achieved in the HiPPO framework but what is remembered will depend on the task it is trained on. In contrast, our model leads to reconstruction guarantees in the observation space, regardless of the task, as shown in our Result 4.1. Tracking the d features of the time series is what allows us to enforce long memorization by design.
> > That said, an avenue for future work that we highlight in the original manuscript is the investigation of how to leverage correlations between the different dimensions of the time series to provide more efficient encoding of the orthogonal polynomial projection coefficients.

---

> > ### Author Response · Authors · 2022-11-18
> > **Answers to reviewer's comments (2/2)**
> >
> > ### About the computational tradeoffs.
> >
> > `Computational tradeoffs of this method and baselines should be discussed and quantitatively compared, particularly as a function of d and N. `
> >
> > As we acknowledge in the Remarks of Section 4.1, the computational complexity of the methods scales with $d \times N$ as we project each temporal dimension individually. To ensure fair comparison, the baselines were run with a varying number of hidden dimensions and the ones corresponding to best validation performance were kept.
> > The reconstruction error does not depend on the number of dimensions but does scale inversely proportionally to N, as shown in our Result 4.1 and in our discussion of it in the manuscript.
> >
> > ### About the S4 model
> >
> > `This issue of blowing up the dimension by a factor of N is in fact one of the main problems addressed by later extensions of HiPPO to full state-space models such as S4. These baselines should be included where appropriate, as they can be used as drop-in models for most of the considered applications (note that they can also be applied to many "irregularly-sampled" time series problems where the sampling is provided as a mask).`
> >
> > We are pleased to inform the reviewer that we have successfully run the S4 model and included the results in the paper. We ran S4 for multiple numbers of hidden dimensions [32,64,128,258,512,1024] and selected the best configuration based on the validation error.  For convenience, we copy an excerpt of these results below. The complete tables are available in the revised version of the paper.
> >
> > |         | Classification | Classification | Classification | Reconstruction | Reconstruction | Reconstruction |
> > |---------|----------------|----------------|----------------|----------------|----------------|----------------|
> > |         | Synthetic      | Lorenz63       | Lorenz96       | Synthetic      | Lorenz63       | Lorenz96       |
> > | S4      | 0.994 +- 0.003 | 0.911 +- 0.005 | 0.948 +- 0.016 | 0.032 +- 0.006 | 0.428 +- 0.040 | 0.171 +- 0.008 |
> > | PolyODE | 0.994 +- 0.003 | 0.992 +- 0.000 | 0.984 +- 0.002 | 0.012 +- 0.002 | 0.034 +- 0.008 | 0.038 +- 0.008 |
> >
> > We observe that S4 is indeed a very solid baseline, which matches PolyODE on the Synthetic Classification. However, S4 still suffers from the same issues as HiPPO, namely, the inability to meaningfully interpolate the data between observations. What is more, as we stated above, S4 (and HiPPO) does not have any time-series reconstruction ability as the orthogonal polynomial projection is performed on the hidden state. While projecting the hidden state to a single dimensional process alleviates the computational burden, it also brings its own disadvantages when it comes to enforcing long term memorization of the time series.
> >
> > ### Concluding remarks about our work and its positioning
> >
> > We hope that the responses above will have convinced the reviewer of the value of our contribution and clarified our positioning.
> >
> > We build upon the HiPPO framework but we differ in several key aspects. *PolyODE is a Neural ODE method* that can leverage the other time series in the dataset to meaningfully interpolate the input process in between irregular observations.
> >
> > What is more, we took the approach of *projecting the observed process onto the basis of orthogonal polynomial*, which allows us to enforce long-range memorization by design.
> >
> > Importantly, the original manuscript compared our model to HiPPO and HiPPO-RNN. Over the rebuttal period, we added additional comparisons to S4 and show that the limitations pointed above result in lower performance compared to our approach. i.e. the application of a powerful discrete time model (applied to data with irregular sampling rates) underperforms a method developed from the ground up for this domain.

---

> > ### Comment · Reviewer_uK5k · 2022-11-21
> > **Response to Author**
> >
> > Thank you for providing a detailed response and clarifications, and providing anonymized code. The clarifications and improvements to the paper together with code have made it more clear that the paper proposes different technical contributions than prior work. However, I still think that the contrast between PolyODE and closely related past work is missing a more detailed comparison, and have several questions about technical details and the relation to HiPPO.
> >
> > ### About tracking each of the d temporal dimensions
> >
> > If I'm interpreting this response correctly, you are saying that HiPPO projects down to dimension 1, which loses the ability to memorize. On the other hand PolyODE tracks all $d$ dimensions independently, which is necessary for memorization. I agree with this, but this seems like an empty argument; HiPPO chose a different tradeoff to focus on (speed rather than perfect construction of the history), and could easily be applied to all $d$ dimensions independently as well. How does PolyODE differ from this?
> >
> > ### About the reconstruction process in PolyODEs
> >
> > I am confused about a more general point about the reconstruction process (Figure 1): are the reconstructions "fixed" operators, or learned from data? If they are fixed, then equation (9) makes sense which is just the HiPPO reconstruction formula. But then how are the NODE reconstructions calculated? On the other hand, you seem to be implying that the NODE methods all *learn* an unsupervised embedding which is used for reconstruction; but then how does equation (9) fit in, which is a fixed operator that doesn't depend on any training?
> >
> > ### HiPPO baselines
> >
> > As per the above discussion, the HiPPO method can trivially be applied on all $d$ dimensions independently (particularly the HiPPO-obs baseline, which doesn't use the RNN part). In the experiments, are they using the "original version" of HiPPO that projects to dimension 1, or is HiPPO-obs being applied on all dimensions independently?
> >
> > ### S4 baselines and model depths
> >
> > Are all models a "1-layer" model? Or are they DNNs with repeated layers? This can make a difference for some baselines, because some methods are non-linear (any of the NODE models, HiPPO-RNN) and some methods are linear (HiPPO-obs, S4). In practice, the latter methods always need to be put into a DNN with non-linearities to obtain enough expressiveness for different tasks. The authors state that "We ran S4 for multiple numbers of hidden dimensions [32,64,128,258,512,1024]", but this seems to imply a single layer model; in the S4 paper, they generally use a deeper model (e.g. 4 layers) with smaller dimension. Also, is this dimension referring to the "model dimension" (the number of independent channels which are being processed each by S4 unit) or the "state dimension" (i.e. $N$, or the HiPPO dimension)?
> >
> >
> > ### Difference to HiPPO
> >
> > The authors make statements of the following form in several places:
> >
> > > Indeed, the HiPPO and S4 framework perform the orthogonal polynomial in the latent space while we perform it on the observation space. This crucial difference leads to reconstruction guarantees of the time series, as our Result 4.1 shows. Because one cannot invert the latent vectors back to the observation space, HiPPO and S4 fail to provide accurate backward reconstruction.
> >
> > I don't quite understand how this differs from HiPPO. What is the HiPPO "latent state"? My understanding of HiPPO is that a given observation is projected onto several orthogonal polynomials, which then allows the original observation to be reconstructed. Backwards reconstruction of the observation is precisely what HiPPO was designed to achieve.
> >
> > > We build upon the HiPPO framework but we differ in several key aspects. PolyODE is a Neural ODE method that can leverage the other time series in the dataset to meaningfully interpolate the input process in between irregular observations.
> >
> > To clarify a consequence of this: if PolyODE is applied to a $d$-dimensional data, and is asked to reconstruct the first dimension, does this depend on the other $d-1$ dimensions? That is, are all $d$ dimensions independent (as would be in a naive application of HiPPO)? If not, how are they being mixed; do the different input dimensions affect each other purely through the NODE? This would also seem like a strange and potentially undesirable property for reconstruction.
> >
> > ### Method description
> >
> > I think I am overall confused by the exact formulation of the method in equation (6). It looks like essentially applying HiPPO directly, with an additional component involving a neural ODE. It's not clear how these two separate components interact:
> > - What is the relation between $h_0$, $h$, and $c$? I.e. which ones influence each other and how?
> > - Which components are being passed through an ODE solver? Figure 2 depicts all of them, but other text contradicts this for $c$ (and to my understanding the HiPPO coefficients should not have to go through a solver).

---

> > > ### Author Response · Authors · 2022-11-23
> > > **Second response to Reviewer uK5k (1/4)**
> > >
> > > Dear Reviewer uK5k,
> > >
> > > Thank you very much for your detailed and swift answer. We are glad to read that our responses have addressed some of your initial concerns including the technical and theoretical distinctions between our work and HiPPO. We hope that the following explanations will help clarify the remaining points.
> > >
> > > ### About tracking each of the d temporal dimensions
> > >
> > > `If I'm interpreting this response correctly, you are saying that HiPPO projects down to dimension 1, which loses the ability to memorize. On the other hand PolyODE tracks all d dimensions independently, which is necessary for memorization. I agree with this, but this seems like an empty argument; HiPPO chose a different tradeoff to focus on (speed rather than perfect construction of the history), and could easily be applied to all d dimensions independently as well. How does PolyODE differ from this?`
> > >
> > > The choice to use a single polynomial rather than multiple ones is a choice embedded into HiPPO architecture and follow ups. Projecting to a 1 dimensional process is thus a structural choice and ablating this requires a different architecture.
> > > Yet, we fully agree that this approach is indeed a very important ablation. Indeed, it actually corresponds to the HiPPO-obs that we compare against in the original manuscript. In this setting, we use the HiPPO framework to memorize the d different longitudinal components of the observed time series.
> > > PolyODE differs from this approach in several ways:
> > >
> > > a) It is able to provide a more meaningful interpolation in between the successive time points, through the incorporation of the NeuralODE part of our model.
> > >
> > > b) While HiPPO-obs would have to do a trapezoidal integration between successive time points, PolyODE provides a continuous input signal that is, crucially, learnt from the dynamics of all the time series available in the dataset. This leads to a more accurate state representation of the dynamical system.
> > >
> > > c) Importantly, HiPPO-obs is *not* capable of forecasting. As it merely memorizes the input process, it does not learn underlying dynamics of the data and can thus not be used for forecasting, which is thus missing an important feature of time series models.
> > >
> > > ### About the reconstruction process
> > >
> > > `I am confused about a more general point about the reconstruction process (Figure 1): are the reconstructions "fixed" operators, or learned from data? If they are fixed, then equation (9) makes sense which is just the HiPPO reconstruction formula. But then how are the NODE reconstructions calculated? On the other hand, you seem to be implying that the NODE methods all learn an unsupervised embedding which is used for reconstruction; but then how does equation (9) fit in, which is a fixed operator that doesn't depend on any training?`
> > >
> > > Reconstructions in PolyODE are fixed operators, as indeed suggested by Equation (9). Classical NODE reconstructions (such as GRU-ODE or ODE-RNN) are computed using backward integration of the hidden process. That is, conditioned on the hidden at time $t$, one can reconstruct the time series at time $t’<t$ using :
> > >
> > > $\hat{\mathbf{x}}(t’) = g(\mathbf{h}(t’)) $
> > >
> > > $\mathbf{h}(t’) = \mathbf{h}(t) + \int_t^{t’} \phi(\mathbf{h}(s)) ds$
> > >
> > > where $\phi$ is the neural network characterizing the ODE.
> > >
> > > We will include this explanation in the revision of the paper.
> > >
> > > ### About HiPPO baselines
> > >
> > > `As per the above discussion, the HiPPO method can trivially be applied on all d dimensions independently (particularly the HiPPO-obs baseline, which doesn't use the RNN part). In the experiments, are they using the "original version" of HiPPO that projects to dimension 1, or is HiPPO-obs being applied on all dimensions independently?`
> > >
> > > As described in our answer above, HiPPO-obs uses the HiPPO framework to track the d longitudinal components of the observed time series. The original manuscript does compare against HiPPO-obs, HiPPO-RNN (which performs a projection to a single dimensional process) and S4.

---

> > > > ### Comment · Reviewer_uK5k · 2022-11-27
> > > > **Response to Authors round 2 (2/2)**
> > > >
> > > > ### Method description
> > > >
> > > > Thank you for the detailed description; the latest response has substantially cleared up my understanding of the method details. To give some direct feedback on the presentation, I think that Section 4.1 was not very helpful for understanding the method (Equations (6) and (7), Figure 2, Algorithm 1, etc.). Possibly it hits the classic pitfall of focusing on formality and technical rigor rather than clarity. I think it's likely that there's a semi-formal description and better figure that would give a much more clear high-level description of the method.
> > > >
> > > > ### Review update
> > > >
> > > > Overall, I think that this paper does contribute new technical ideas that improve on baselines on addressing the problems it is concerned about (e.g. irregular sampled time series). However, I think the method description is incredibly confusing, which also partly stems from somewhat inaccurate portrayals of prior work that makes the method even more instead of less confusing for experts in the space. I have updated my original review and my current score is borderline; while I would like to give the benefit of the doubt, I'm not sure whether the presentation of the paper could be improved enough between the current version and a hypothetical camera ready, which (in my opinion) would require substantial changes.

---

> > > > ### Comment · Reviewer_uK5k · 2022-11-27
> > > > **Response to Authors round 2 (1/2)**
> > > >
> > > > Thanks again for providing detailed responses! I have a few more comments and questions, and have updated my score again.
> > > >
> > > > ### Reconstruction process
> > > >
> > > > This response clears it up better. I personally didn't find this obvious in the paper/figure and think it is worth putting somewhere, at least in the Appendix. Also, there's still a detail I am not familiar with: when are neural ODEs backwards reconstructible? I assume that there are some conditions on the ODE. I guess it seems sufficient that the ODE depends only on the hidden state $\mathbf{h}((t))$ and not directly on the time $t$ as this then defines a flow; is this an established condition in the NODE literature?
> > > >
> > > > Since PolyODE uses a NODE component as well as the HiPPO component, does it require the NODE backwards integration as well for reconstruction? Or is only the HiPPO part used?
> > > >
> > > > Since reconstruction of the backwards process is an important part of this paper, I feel like these details should be more transparent with discussion/references where appropriate.
> > > >
> > > > ### S4 baseline
> > > >
> > > > Thanks for the clarification. It was probably not necessary to search over the state dimension as the S4 papers say they always used a fixed dimension; I was just checking some details of the baselines, which might be worth clarifying in experimental details as well (probably Appendix).
> > > >
> > > > ### Difference from HiPPO
> > > >
> > > > > The choice to use a single polynomial rather than multiple ones is a choice embedded into HiPPO architecture and follow ups. Projecting to a 1 dimensional process is thus a structural choice and ablating this requires a different architecture.
> > > >
> > > > I will disagree with this as it is a trivial distinction, and the original HiPPO architecture was probably chosen for computational reasons. I forgot to respond to this in the last post, but I think that the computational tradeoffs of PolyODE in practice could be much more transparent in the paper. For example a more detailed comparison of HiPPO-RNN vs HiPPO-obs vs PolyODE would clarify the tradeoffs involved much better. Additionally, the follow-ups to HiPPO (LSSL and S4) apply independent HiPPO across $d$ different components of the observed time series so that claim seems false (but I agree they differ from PolyODE in other ways). I do agree with your other responses (a) (b) (c) in this section.
> > > >
> > > > > The difference between the HiPPO models and our frameworks here lies in what is the input signal that is projected onto the basis of orthogonal polynomials. In the HiPPO RNN paper, the hidden state of a recurrent architecture is mapped to a single dimension which is then projected onto the orthogonal basis via the HiPPO matrix.
> > > >
> > > > I don't think this is accurate at all. If I'm interpreting this correctly, this seems to be making an artificial distinction between *HiPPO* and *HiPPO-RNN*. HiPPO-RNN is just a thin (perhaps not very well-designed, to be honest) wrapper around HiPPO. HiPPO is the core component which is what PolyODE is using as a black box and is doing the "reconstruction of observation" part.
> > > >
> > > > Overall, I suspect many of our misunderstandings are because I think that the comparison should be about *the technical core of prior work* (e.g. the HiPPO ODE), not *the particular way it was applied in practice* (e.g. HiPPO-RNN, 1D inputs, etc.). I don't think it is accurate at all to distinguish PolyODE from prior work with descriptions such as "Indeed, the HiPPO framework perform the orthogonal polynomial in the latent space while we perform it on the observation space", which frequently appear in the paper. The reason PolyODE has its reconstruction capabilities is because prior work *does have this ability as well* and PolyODE is *using that method as a black box*, so this seems like a pretty large mischaracterization of prior work. I do acknowledge that some effort was made to bridge the gap (e.g. the HiPPO-obs baseline), but this should be fleshed out better and I think that many of the descriptions of prior work in this paper are misrepresented.
> > > >
> > > > > Clarification: when we write that PolyODE is able to leverage other time series in the dataset, we mean other realizations of d-dimensional time series. In the MIMIC example, PolyODE makes use all other patients in the data to interpolate the time series in between the observations.
> > > >
> > > > I still don't quite understand what you mean by this. Where is the explicit dependence on the other training examples (i.e. the other time series)? This response seems to indicate that PolyODE is a *non-parametric method*, which goes against what I understood previously. Or is this essentially saying that since HiPPO is a fixed operator, it does not use the actual data; whereas since PolyODE has a learnable NODE component, it is implicitly learning from all training examples to perform the interpolation? If the former, I'm once again confused about what PolyODE is doing; if the latter, I think this is not a very good description as it applies to all parametric methods.

---

> > > ### Author Response · Authors · 2022-11-23
> > > **Second response to Reviewer uK5k (2/4)**
> > >
> > > ### S4 baselines and model depths
> > >
> > > `Are all models a "1-layer" model? Or are they DNNs with repeated layers? This can make a difference for some baselines, because some methods are non-linear (any of the NODE models, HiPPO-RNN) and some methods are linear (HiPPO-obs, S4). In practice, the latter methods always need to be put into a DNN with non-linearities to obtain enough expressiveness for different tasks.
> > > The authors state that "We ran S4 for multiple numbers of hidden dimensions [32,64,128,258,512,1024]", but this seems to imply a single layer model; in the S4 paper, they generally use a deeper model (e.g. 4 layers) with smaller dimension. Also, is this dimension referring to the "model dimension" (the number of independent channels which are being processed each by S4 unit) or the "state dimension" (i.e. N, or the HiPPO dimension)?`
> > >
> > > We followed the recommendations of the S4 authors and used a 4 layers model in our experiments. The model therefore indeed includes non-linearities. The hidden dimension we have used refers to the model dimension.
> > >
> > > For completeness, we have run an additional hyper-parameters search with the state-dimension as well (so searching over both the model and state dimension). Again, the best hyper-parameters were selected based on validation performance. The results can be found below. S4-extended corresponds to the results of that extended hyper-parameters search. We observe a slight improvement on the Synthetic dataset but the results remain within the same confidence intervals for the other datasets. The same trend can be observed for the other irregular rates (we will add these results to Tables 4,5, and 6). S4-extended here refers to the new, extended, hyper-parameters search while S4 corresponds to the previous one.
> > >
> > > |             | Classification | Classification | Classification | Reconstruction | Reconstruction | Reconstruction |
> > > |-------------|----------------|----------------|----------------|----------------|----------------|----------------|
> > > |             | Synthetic      | Lorenz63       | Lorenz96       | Synthetic      | Lorenz63       | Lorenz96       |
> > > | S4          | 0.994 +- 0.003 | 0.911 +- 0.005 | 0.948 +- 0.016 | 0.032 +- 0.006 | 0.428 +- 0.040 | 0.171 +- 0.008 |
> > > | S4-extended | 0.995 +- 0.002 | 0.909 +- 0.015 | 0.937 +- 0.018 | 0.017 +- 0.004 | 0.439 +- 0.026 | 0.208 +- 0.066 |
> > > | PolyODE     | 0.994 +- 0.003 | 0.992 +- 0.000 | 0.984 +- 0.002 | 0.012 +- 0.002 | 0.034 +- 0.008 | 0.038 +- 0.008 |
> > >
> > >
> > > We are happy to run additional experiments with other sets of hyper-parameters that you would consider more appropriate.

---

> > > ### Author Response · Authors · 2022-11-23
> > > **Second response to Reviewer uK5k (3/4)**
> > >
> > > ### About the difference with HiPPO
> > >
> > > `I don't quite understand how this differs from HiPPO. What is the HiPPO "latent state"? My understanding of HiPPO is that a given observation is projected onto several orthogonal polynomials, which then allows the original observation to be reconstructed. Backwards reconstruction of the observation is precisely what HiPPO was designed to achieve.`
> > >
> > > The HiPPO matrix is designed to define a state space model that corresponds to projecting an input signal onto a basis of orthogonal polynomials in order to “memorize” it.
> > >
> > > The difference between the HiPPO models and our frameworks here lies in what is the input signal that is projected onto the basis of orthogonal polynomials. In the HiPPO RNN paper, the hidden state of a recurrent architecture is mapped to a single dimension which is then projected onto the orthogonal basis via the HiPPO matrix. In the S4 paper, it is a non-invertible mapping of the input that is projected. In both cases, the input signal that is being memorized is not the observed process but a non-invertible transformation thereof.
> > >
> > > This choice is valid for the applications showcased in these papers. Indeed, these papers focus on direct classification or regression of long time series. In this case, projecting the hidden space allows to learn what information is worth remembering for the specific task they aim at solving. But, as a consequence, HiPPO RNN and S4 do not explicitly impose any reconstruction ability on the input process, due to the non-invertibility between the signal that is projected and the observed process.
> > > In contrast, our approach aims at providing a general informative embedding of time series with Neural ODEs that provably enforce long-term memory of the observed process. This consideration leads us to projecting the input process onto the basis of orthogonal polynomials rather than the hidden process (as in HiPPO or S4).
> > >
> > > `If PolyODE is applied to a d-dimensional data, and is asked to reconstruct the first dimension, does this depend on the other d−1 dimensions? That is, are all d dimensions independent (as would be in a naive application of HiPPO)? If not, how are they being mixed; do the different input dimensions affect each other purely through the NODE? This would also seem like a strange and potentially undesirable property for reconstruction.`
> > >
> > > Clarification: when we write that PolyODE is able to leverage other time series in the dataset, we mean other realizations of d-dimensional time series. In the MIMIC example, PolyODE makes use all other patients in the data to interpolate the time series in between the observations.
> > >
> > > As for the reconstruction, only the coefficients of each dimension j ($c_n^j(t)$)  are required for the reconstruction (as shown in Equation (9)). Nevertheless, the interpolating signal used to compute the coefficients depends on all the other dimensions, through the Neural ODE. Hence, the reconstruction of dimension $j$ indirectly depends on the other dimensions.
> > > We are not convinced that this should be considered a potentially undesirable property. This allows for correlations between the different dimensions of the input signals to be utilized, and thus improve the representation abilities of the model.

---

> > > ### Author Response · Authors · 2022-11-23
> > > **Second response to Reviewer uK5k (4/4)**
> > >
> > > ### About the method description
> > >
> > > `I think I am overall confused by the exact formulation of the method in equation (6). It looks like essentially applying HiPPO directly, with an additional component involving a neural ODE. It's not clear how these two separate components interact:
> > > What is the relation between h0, h, and c? I.e. which ones influence each other and how?`
> > >
> > > The interactions between these different processes differ depending on the regime of the model.
> > >
> > > We distinguish two regimes : the integration step (that takes place in between observations) and the update step (that takes place at the times of observations). Note that iterating between integration regime and update regime is standard in many classical architectures (e.g. Kalman filters) and Neural ODEs architectures (e.g. ODE-RNN).
> > >
> > >
> > > - During the **integration step** (Eq.6):
> > >
> > > During the integration step, we integrate the hidden process $\mathbf{h}(t)$ and the coefficients $\mathbf{c}(t)$ forward in time until the next observation.
> > >
> > > The hidden process $\mathbf{h}(t)$ influences itself through the Neural ODE parametrized by $\phi_{\theta}$ (Eq.6). As $\phi_{\theta}$ is a fully connected MLP,  all dimensions of $\mathbf{h}(t)$ can be impacted by any other (the dynamical system if fully coupled). As such, $\mathbf{h}_0(t)$ both influence and are influenced by the whole $\mathbf{h}(t)$. This allows to harness correlations between different dimensions in the time series to produce a more accurate interpolation.
> > >
> > > The coefficients $\mathbf{c}(t)$ are influenced by the values of $\mathbf{h}(t)$ through $\mathbf{h}_0(t)$ only. $\mathbf{h}_0(t)$ provides the signal that will be memorized by projecting onto the orthogonal polynomial basis. The $\mathbf{c}(t)$ serve as memory banks and do not influence the dynamics of $\mathbf{h}(t)$ during the integration step.
> > >
> > > - During the **update** step (Eq.7.):
> > >
> > > The update step serves the role of incorporating new observations in the hidden representation of the system. It proceeds by
> > >
> > > (1) reinitializing the hidden states of the system with the orthogonal polynomial projection coefficients $\mathbf{c}(t)$. We set $\mathbf{h}_j(t) := \mathbf{c}^j(t)$.
> > >
> > > (2) for hidden state 0 ($\mathbf{h}_0(t)$), which corresponds to the observation space, we reset it to the observed values. That is, $\mathbf{h}_0(t)_j := x_j(t)$
> > >
> > > The following integration step will then predict how the observations would evolve over time until the next available observation.
> > >
> > > `Which components are being passed through an ODE solver? Figure 2 depicts all of them, but other text contradicts this for c (and to my understanding the HiPPO coefficients should not have to go through a solver).`
> > >
> > > All components $\mathbf{h}(t)$ and $\mathbf{c}(t)$ are passed through an ODE solver, as suggested by Equation 6. The whole system of Equation 6 corresponds to the ODE that we integrate numerically. Notably, the projection coefficients are also passed through the solver, using the HiPPO relation:
> > >
> > > $\frac{d\mathbf{c}(t)}{dt} =A_{\mu} \mathbf{c}(t) + B_{\mu} f(t)$
> > >
> > > The original HiPPO and S4 papers also implicitly use a numerical solver while deriving their recurrence equation. Indeed, they use the trapezoidal rule, as we discussed in our previous answer.

---

> ### Author Response · Authors · 2022-12-06
> **Third response to reviewer uK5k (1/3)**
>
> We want to deeply thank reviewer uK5K for its engagement and very pertinent feedback that significantly helped make our paper stronger. We are glad to read that the discussion period has helped resolve some uncertainty about certain aspects of our work which were not described clearly enough. We have now incorporated the results of our fruitful exchange in the paper.
>
> While the update period for the manuscript is over, we have included excerpts of the paper where significant changes have been made. We hope this will help convince reviewer uk5K of the improved clarity of our method’s description. We also answer remaining comments directly below.
>
> Thank you again,
>
> The authors
>
> ## Reconstruction process
>
> `This response clears it up better. I personally didn't find this obvious in the paper/figure and think it is worth putting somewhere, at least in the Appendix. Also, there's still a detail I am not familiar with: when are neural ODEs backwards reconstructible? I assume that there are some conditions on the ODE. I guess it seems sufficient that the ODE depends only on the hidden state h((t)) and not directly on the time t as this then defines a flow; is this an established condition in the NODE literature?`
>
> This reconstruction exists and is unique if $\phi$ is continuous in $t$ and Lipschitz continuous in $\mathbf{h}$, according to the Picard-Lindelöf theorem (Nagle et al. 2011). As neural networks parametrized with continuous activation functions (\emph{e.g.} hyperbolic tangent or sigmoid) are Lipschitz continuous, we have that the above reconstruction exists and is unique for these activation functions.
>
> `Since PolyODE uses a NODE component as well as the HiPPO component, does it require the NODE backwards integration as well for reconstruction? Or is only the HiPPO part used?`
>
> Only the HiPPO part is used.
>
> `Since reconstruction of the backwards process is an important part of this paper, I feel like these details should be more transparent with discussion/references where appropriate.`
>
> We have now added more details about the reconstruction process in Appendix E.
>
> **Excerpt from Appendix E :**
>
> Reconstructions in PolyODE are fixed operators, as suggested by Equation 3.1. The reconstruction is obtained by using the orthogonal polynomial coefficients at a particular step. This procedure ensures accurate backward prediction performance, as supported by Result 4.1.
>
> In contrast, classical NODE reconstructions (such as GRU-ODE or ODE-RNN, and depicted in Figure 1) are computed using backward integration of the hidden process. That is, conditioned on the hidden at time , one can reconstruct the time series at time using :
>
>
>  $\hat{\mathbf{x}}(t') = g(\mathbf{h}(t'))$
>
>  $\mathbf{h}(t') = \mathbf{h}(t) + \int_{t}^{t'} \Phi(\mathbf{h}(s)) ds$
>
>
> where $\phi$ is the neural network characterizing the NODE.
>
> We note that this reconstruction exists and is unique if $\phi$ is continuous in $t$ and Lipschitz continuous in $\mathbf{h}$, according to the Picard-Lindelöf theorem (Nagle et al. 2011). As neural networks parametrized with continuous activation functions (\emph{e.g.} hyperbolic tangent or sigmoid) are Lipschitz continuous, we have that the above reconstruction exists and is unique for these activation functions.
>
> R. K. Nagle, E. B. Saff, and A. D. Snider.Fundamentals of differential equations and boundary value problems. Pearson education, 2011.

---

> ### Author Response · Authors · 2022-12-06
> **Third response to reviewer uK5k (2/3)**
>
> ## Differences with HiPPO
>
> `I don't think this is accurate at all. If I'm interpreting this correctly, this seems to be making an artificial distinction between HiPPO and HiPPO-RNN. HiPPO-RNN is just a thin (perhaps not very well-designed, to be honest) wrapper around HiPPO. HiPPO is the core component which is what PolyODE is using as a black box and is doing the "reconstruction of observation" part.`
>
> We agree with this description. HiPPO-RNN acts as a wrapper that will apply HiPPO on the hidden state of an RNN. As our approach uses the HiPPO operator on the observation space, it makes sense to compare it with an HiPPO operator that applies on the observed process (HiPPO-obs).
>
> `I do acknowledge that some effort was made to bridge the gap (e.g. the HiPPO-obs baseline), but this should be fleshed out better and I think that many of the descriptions of prior work in this paper are misrepresented.`
>
> We want to thank the reviewer again for pointing this out and we have now improved the clarity with regard to our dependence on the HiPPO framework. We hope that this makes our dependence on HiPPO and our exact contribution much more evident.
>
> **Excerpt from the contributions :**
>
> Point 1 : “We propose a novel dynamics function for a neural ODE resulting in PolyODE, a model that learns a global representation of high-dimensional time series and is capable of long-term forecasting and reconstruction by design. PolyODE is the first investigation of the potential of the HiPPO operator for neural ODEs architectures.”
>
> **Excerpt from related works :**
>
> Our method builds upon the HiPPO framework which defines an operator to compute the coefficients of the projections on a basis of orthogonal polynomials. HiPPO-RNN and S4 are the most prominent examples of architectures building upon that framework [1,2].
>
> These architectures rely on a linear interpolation of the data in between observations, which can lead to a decrease of performance when the sampling rate of the input process is low. Furthermore, HiPPO-RNN and S4 perform the orthogonal polynomial projection of a non-invertible representation of the input data, which therefore doesn't enforce reconstruction in the observation space by design.
>
> This design choice has been motivated by the fact that these approaches focus on long term dependency for a single target task (such as trajectory classification). In contrast, this work aims at exploring the abilities of the HiPPO operator for representation learning of irregular time series, when the downstream task is not known in advance. To our knowledge, PolyODE is also the first work that investigates the advantages of the HiPPO operator in the context of neural ODE architectures.
>
> [1] A. Gu et al. Hippo:Recurrent memory with optimal polynomial projections.
> [2] A. Gu et al. Efficiently modeling long sequences with structured state spaces.
>
> `I still don't quite understand what you mean by this. Where is the explicit dependence on the other training examples (i.e. the other time series)? This response seems to indicate that PolyODE is a non-parametric method, which goes against what I understood previously. Or is this essentially saying that since HiPPO is a fixed operator, it does not use the actual data; whereas since PolyODE has a learnable NODE component, it is implicitly learning from all training examples to perform the interpolation? If the former, I'm once again confused about what PolyODE is doing; if the latter, I think this is not a very good description as it applies to all parametric methods.`
>
> We indeed mean the latter here : “PolyODE has a learnable NODE component, it is implicitly learning from all training examples to perform the interpolation”. We agree this is a very general statement. Yet, as you state above, HiPPO is a fixed operator and thus needs to be embedded in a larger architecture to use the information contained in multiple time series. PolyODE can thus be seen as such a larger architecture, based on Neural ODEs, that learns from the whole dataset and embeds an HiPPO operator.

---

> ### Author Response · Authors · 2022-12-06
> **Third response to reviewer uK5k (3/3)**
>
> ## Method description
>
> `I think that Section 4.1 was not very helpful for understanding the method (Equations (6) and (7), Figure 2, Algorithm 1, etc.). Possibly it hits the classic pitfall of focusing on formality and technical rigor rather than clarity. I think it's likely that there's a semi-formal description and better figure that would give a much more clear high-level description of the method.`
>
> We have updated the text in the paper according to our discussion. In particular, we have extended the description of the integration and update steps:
>
> **Excerpt preamble 4.1:**
>
> Similarly as classical filtering architectures (e.g. Kalman filters and ODE-RNN), PolyODE alternates between two regimes : an integration step (that takes place in between observations) and an update step (that takes place at the times of observations), described below.
>
> **Excerpt Integration step:**
>
> During the integration step, we integrate both the hidden process $\mathbf{h}(t)$ and the coefficients $\mathbf{c}(t)$ forward in time, using the system of Equation 6. At each time step, we can provide an estimate of the time series $\hat{\mathbf{x}}(t)$ conditioned on the hidden process $\mathbf{h}(t)$, with $\hat{\mathbf{x}}(t) = g(\mathbf{h}(t))$.
>
> The coefficients $\mathbf{c}(t)$ are influenced by the values of $\mathbf{h}(t)$ through $\mathbf{h}_0(t)$ only. The process $\mathbf{h}_0(t)$ provides the signal that will be memorized by projecting onto the orthogonal polynomial basis.
> The $\mathbf{c}(t)$ serve as memory banks and do not influence the dynamics of $\mathbf{h}(t)$ during the integration step.
>
> **Excerpt Update step:**
>
> The update step serves the role of incorporating new observations in the hidden representation of the system. It proceeds by
>
> (1) reinitializing the hidden states of the system with the orthogonal polynomial projection coefficients $\mathbf{c}(t)$: $\mathbf{h}_j(t_i) = \mathbf{c}^j(t_i)$; and
>
> (2) resetting  $\mathbf{h}_0 (t)$  to the newly collected observation: $\mathbf{h}_0(t_i) = \mathbf{x}_i(t)$

---

> > ### Comment · Reviewer_uK5k · 2022-12-11
> > **Response**
> >
> > I appreciate the use of excerpts despite being unable to modify the draft. I'll respond briefly to the latest round of responses.
> >
> > > We have updated the text in the paper according to our discussion. In particular, we have extended the description of the integration and update steps:
> >
> > This high-level description is much more helpful for understanding the intuition and the role each of the components $h(t), h_0(t), c(t)$ play.
> >
> > > We agree this is a very general statement. Yet, as you state above, HiPPO is a fixed operator and thus needs to be embedded in a larger architecture to use the information contained in multiple time series. PolyODE can thus be seen as such a larger architecture, based on Neural ODEs, that learns from the whole dataset and embeds an HiPPO operator.
> >
> > I would make a slight quibble that this is conflating the "HiPPO ODE" and "architectures based on HiPPO" (e.g. HiPPO-RNN, S4) again, but in the other direction from the earlier ones. But it's overall a minor point as long as the comparison is clear.
> >
> > > Reconstruction process
> >
> > Thanks for the clarification!
> >
> > ### Update
> >
> > Overall, I commend the authors for making significant changes to the paper throughout the rebuttal process that I do think has improved the presentation significantly. I have updated my score and am willing to argue for the paper.

---

### Official Review · Reviewer_mcs4 · 2022-10-28

**Confidence:** 4
**Correctness:** 3
**Technical Novelty And Significance:** 2
**Empirical Novelty And Significance:** 3
**Recommendation:** 6

**Clarity, Quality, Novelty And Reproducibility:**

- Quality: Good
- Clarity: Good
- Originality: Fair

**Strength And Weaknesses:**

Strength:
- The idea of using orthogonal polynomials to solve long-range properties of time series with continuous-time models is interesting.
- The paper shows some theoretical results as well as competitive performance of the proposed model.

Weakness:
- As one of the main contributions, the paper includes the way to use adaptive solvers. However, the paper just briefly mentions Backward Euler and Adams-Moultons [Sauer, 2011]. Firstly, it is not self-contained. Many readers from machine learning background don’t know about such numerical methods. Therefore, the current explanation does not provide . Secondly, although the observation about the stiffness of ODEs is interesting, using such solvers is not a significant contribution.
- Missing comparisons with neural controlled differential equations with log-signature [Morril et al., 2021]. I think this is the main baseline in terms of capturing long-range dynamics. The paper mentions this reference but does not give empirical comparisons and a detailed discussion.
- The stiffness of ODE is elucidated clearly. However, there are no illustrations of the spectral information of the matrix $A$.


**Summary Of The Paper:**

The paper presents neural ordinary differential equations (Neural ODEs) to learn representations of time-series in a latent space. The dynamic of such a latent space uses continuous-time neural ODEs, extending from existing discrete-time versions [Gu et al, 2020] based on orthogonal polynomials. The paper shows that the learned representations can capture long-range memory with theoretical results and empirical results via reconstructing the past of time series.

**Summary Of The Review:**

This is a solid paper even though it extends from Gu et al. 2020 into neural ODE. I lean toward accepting the paper. At the moment, I give a score of 6.

---

> ### Author Response · Authors · 2022-11-18
> **Answers to reviewer's comments**
>
> We warmly thank reviewer mcs4 for their encouraging review which helped us strengthen our paper. Please find our answers below and we will happily provide any complementary information that could help elucidate remaining questions.
>
> ### About numerical integration
> `As one of the main contributions, the paper includes the way to use adaptive solvers. However, the paper just briefly mentions Backward Euler and Adams-Moultons [Sauer, 2011]. Firstly, it is not self-contained. Many readers from machine learning background don’t know about such numerical methods. Therefore, the current explanation does not provide . Secondly, although the observation about the stiffness of ODEs is interesting, using such solvers is not a significant contribution.`
>
> We welcome this suggestion and have now added a brief description of several numerical integration methods in Appendix H (referenced in Section 4.2 in the main paper). Specifically, we described the Euler method, the Dopri-5 method and the Adams-Moulton method. To complement our experiments, we have added a comparison of these solvers both in terms of computation time and in terms of performance (validation loss on the forecasting task). We show that the Euler method is unstable and leads to very large integration error. On the other hand, the Dopri-5 method, that can adjust the step size to keep the error under control, is prohibitively expensive (an order of magnitude slower than Euler and Adams-Moulton).
>
>  Finally, we also added details about the stability of the system of Neural ODE, which explains these experimental findings. We report the spectral signature of the matrix $A_{\mu}$ and show that its stiffness ratio grows with the number of projection coefficients $N$.
> Our contribution in the analysis here is not in the development of any novel solvers but rather an ablation to showcase how the choice of right solver can take a method from being passably functional to outperforming many state of the art methods on forecasting and reconstruction.
>
> `The stiffness of ODE is elucidated clearly. However, there are no illustrations of the spectral information of the matrix A.`
>
> As stated above, we added details about the stability of the system of Neural ODE in Appendix H. We report the spectral signature of the matrix $A_{\mu}$ and show that its stiffness ratio grows with the number of projection coefficients $N$.
>
>
> ### About Neural Rough Differential Equations
>
> `Missing comparisons with neural controlled differential equations with log-signature [Morril et al., 2021]. I think this is the main baseline in terms of capturing long-range dynamics. The paper mentions this reference but does not give empirical comparisons and a detailed discussion.`
>
> We have now added the comparison with the neural rough differential equations (RDE) [Moorril et al., 2021]. As with the other baselines, we have used hyper-parameter tuning of the hidden dimension on the validation set. Unfortunately, we observe that this model does not perform as well as the other competitors.
>
> We conjecture that this is due to the lack of forecasting ability of Neural RDEs ( of CDEs in general). Neural RDEs aim at improving the computational complexity of processing long time series rather than capturing long range dependencies or enforcing long-range memory. While these methods are capable of classification and regression, we found it challenging to extend them to the forecasting scenario (our evaluation scheme relies on forecasting to produce the time series embedding that will be used for the downstream tasks).

---

### Official Review · Reviewer_1xor · 2022-10-31

**Confidence:** 4
**Correctness:** 4
**Technical Novelty And Significance:** 3
**Empirical Novelty And Significance:** 3
**Recommendation:** 6

**Clarity, Quality, Novelty And Reproducibility:**

$\cdot$ This paper is very well organized and clearly written.

$\cdot$ The proposed method is novel and this has been addressed in the above comments.

$\cdot$ The reproducibility is feasible.

**Details Of Ethics Concerns:**

There is no ethics concern in this paper.

**Strength And Weaknesses:**

The strengths of this paper are:

$\mathbf{1}.$ The proposed method is novel and with solid contribution. At the same time, there is a complete presentation of theoretical base, where authors propose the novel dynamic function for PolyODE with comprehensive analysis. It could be a great addition to Neural ODE method in the field.

$\mathbf{2}.$ The empirical experiment is comprehensive with very detailed explanation in experiment setup and scenarios. The choice of data, baseline and design of the experiments are both very appropriate and/adequate

**Summary Of The Paper:**

This paper proposes a neural ODE based method to predict time series data. This method mainly projects the long-term trajectory onto basis of polynomials.

**Summary Of The Review:**

This paper proposes a novel Neural ODE method which leverages a novel dynamic functions. The proposed method is well supported by the theoretical analysis and empirical experiment.

---

> ### Author Response · Authors · 2022-11-18
> **Answers to reviewer's comments**
>
> We warmly thank Reviewer 1xor for their encouraging reviews. We are happy to report that we have implemented the following changes in the manuscript.
>
> - We have added two new baselines : (1) S4, an efficient  state space model implementation that also builds upon the HiPPO framework (2) Neural Rough Differential Equations (RDE) which are an adaptation of controlled differential equations for long time series. While competitive, these methods fail to outperform PolyODE.
>
> - We have clarified the positioning of the paper. In particular, we stressed that our method is a Neural ODE method that solves issues identified with previous Neural ODEs implementation and we have made our connection to the HiPPO framework more apparent.
>
> - We have added an in-depth analysis of the numerical integration of our method. In Appendix H, we described three different numerical integration methods and compare their performance in terms of computation time and validation error. We then analyze the spectral properties of matrix $A_{\mu}$.
>
> We are happy to provide any complementary information that the reviewers may see fit.

---

### Author Response · Authors · 2022-12-06
**Summary of the modifications brought to the paper during the rebuttal period**

Dear reviewers,

We want to thank you again for the insightful feedback and comments that have helped us make our paper better.
We have provided significant changes to the original version to include your suggestions and would like to briefly summarize the modifications we made here.

- We greatly clarified our dependence on the HiPPO framework and our contributions. We have explicitly highlighted the difference in computational/scaling/representational tradeoffs made by related work including SSM and S4 relative to our own work. We described our method to make the HiPPO dependency evident and give a more in-depth description of the previous HiPPO-like architectures. In contrast to these previous works, we stress that our method focuses on :

1. Integrating the HiPPO operator in Neural ODE architectures and investigating its impact on representation learning for time series.

2. Addressing irregular time series with low sampling rate by using a Neural ODE coupled with a continuous time HiPPO operator.

- We improved the description of our method to make it more reproducible and easily understandable. We added new explanations for the integration step, the update step and the reconstruction process.

- We compared our approach against two new baselines:  S4 and Neural RDEs. We showed that, while being competitive, they were not matching the performance of PolyODE on the reconstruction and downstream tasks.

- We included an in-depth analysis of the numerical integration aspects. We now discuss and compare different solvers and study the stability properties of our system of differential equations.

We stay at your disposal for answering any further comment you may have on our work.

Best Regards,

The authors

---

### Decision · Program_Chairs · 2023-01-20

**Decision:**

Accept: poster

**Justification For Why Not Higher Score:**

See weaknesses above.

**Justification For Why Not Lower Score:**

Reviewers achieved consensus on acceptance. The proposed NeuralODE/HiPPO combination is novel enough. See strengths above.

**Metareview: Summary, Strengths And Weaknesses:**

Summary: This paper proposes a blend of two approaches for modeling irregular time series data: it is assumed that a latent space dynamics follows a Neural ODE, and the latent state can be decoded into observations; meanwhile, to retain "long-term memory", another system of ODEs evolve the coefficients of a polynomial approximation to the latent dynamics, a construction borrowed from HiPPO/S4 lines of work.

Strengths:` Building a long-term memory mechanism into Neural ODE based time series models via orthogonal polynomial projections is a novel combination that appears to pan out in the empirical results.

Weaknesses: Mainly around presentation and technical clarity as detailed in the reviews. Some proposals are made to increase coverage of baselines to compare against. Computational tradeoffs of this method and baselines should be discussed and quantitatively compared.

**Note From Pc:**

if the above contains the word "oral" or "spotlight" please see: "oral" presentation means -> notable-top-5% and "spotlight" means -> notable-top-25%. As stated in our emails, we are disassociating presentation type from AC recommendations